# A Comprehensive Molecular and Clinical Investigation of Approved Anti-HCV Drugs Repurposing against SARS-CoV-2 Infection: A Glaring Gap between Benchside and Bedside Medicine

**DOI:** 10.3390/vaccines11030515

**Published:** 2023-02-22

**Authors:** Sneha Bansode, Pawan Kumar Singh, Meenakshi Tellis, Anita Chugh, Narendra Deshmukh, Mahesh Gupta, Savita Verma, Ashok Giri, Mahesh Kulkarni, Rakesh Joshi, Dhruva Chaudhary

**Affiliations:** 1CSIR-National Chemical Laboratory, Biochemical Sciences Division, Dr. Homi Bhabha Road, Pune 411008, India; 2Pandit Bhagwat Dayal Sharma Post Graduate Institute of Medical Sciences, Rohtak 124001, India; 3INTOX Private Limited, Pune 412115, India; 4Academy of Scientific and Innovative Research (AcSIR), Ghaziabad 201002, India

**Keywords:** antiviral, COVID-19, daclatasvir, ledipasvir, sofosbuvir, SARS-CoV-2

## Abstract

The limited availability of effective treatment against SARS-CoV-2 infection is a major challenge in managing COVID-19. This scenario has augmented the need for repurposing anti-virals for COVID-19 mitigation. In this report, the anti-SARS-CoV-2 potential of anti-HCV drugs such as daclatasvir (DCV) or ledipasvir (LDP) in combination with sofosbuvir (SOF) was evaluated. The binding mode and higher affinity of these molecules with RNA-dependent-RNA-polymerase of SARS-CoV-2 were apparent by computational analysis. In vitro anti-SARS-CoV-2 activity depicted that SOF/DCV and SOF/LDP combination has IC_50_ of 1.8 and 2.0 µM, respectively, comparable to remdesivir, an approved drug for COVID-19. Furthermore, the clinical trial was conducted in 183 mild COVID-19 patients for 14 days to check the efficacy and safety of SOF/DCV and SOF/LDP compared to standard of care (SOC) in a parallel-group, hybrid, individually randomized, controlled clinical study. The primary outcomes of the study suggested no significant difference in negativity after 3, 7 and 14 days in both treatments. None of the patients displayed any worsening in the disease severity, and no mortality was observed in the study. Although, the post hoc exploratory analysis indicated significant normalization of the pulse rate showed in SOF/DCV and SOF/LDP treatment vs. SOC. The current study highlights the limitations of bench side models in predicting the clinical efficacy of drugs that are planned for repurposing.

## 1. Introduction

The coronavirus disease 2019 (COVID-19) caused by severe acute respiratory syndrome coronavirus-2 (SARS-CoV-2) resulted in an unprecedented pandemic and continues to be a severe health concern in various parts of the world. As of December 2022, nearly 550 million people have been infected with SARS-CoV-2, resulting in over 6.2 million deaths [https://COVID19.who.int/ (accessed on 28 December 2022)]. The COVID-19 pandemic has enormously increased the burden on the healthcare system worldwide and continues to ravage developing countries even more so. Therefore, reducing the infection and hospitalization rate has become the primary goal [1]. Vaccines from several companies effectively provide immunization and reducing the lethal effects of infection. Short of remdesivir and recently evaluated Ensitrelvir, no other antiviral has been demonstrated to have significant anti SARS-CoV-2 activity [2,3]. This highlights the need to discover and establish a new treatment regime for the management of acute infection. Repurposing the existing anti-viral molecules has been considered one of the strategies for managing COVID-19. Repurposing of approved medicines takes advantage of the established safety profiles, optimized production process, and availability of these molecules in the market. Thus, clinicians and researchers worldwide have been investigating the repurposing of several molecules to treat COVID-19 including anti-malarial molecules such as hydroxychloroquine and anti-viral drugs such as lopinavir, ritonavir, molnupiravir and favipiravir, with sub-optimal results [4,5]. However, as demonstrated by multiple studies, the safety and efficacy of these molecules remain debatable [6].

SARS-CoV-2 is a single-stranded positive-sense RNA virus, and similar to other RNA viruses its genomic replication is catalyzed by ORF1ab, also known as RNA-dependent-RNA-polymerase (RdRp) [7]. Thus, existing anti-viral molecules targeting RdRp may have a higher possibility of being repositioned to treat SARS-CoV-2 infection. Along similar lines, Hepatitis C virus (HCV) and SARS-CoV-2 share a near-identical mechanism for genome replication; hence, the drugs against HCV have the potential for investigation for COVID-19 treatment [8]. Sofosbuvir (SOF), daclatasvir (DCV), and ledipasvir (LDP) are FDA-approved anti-HCV drugs and have been explored for SARS-CoV-2 treatment [9,10]. SOF blocks the action of RdRp, while DCV and LDP are inhibitors of Nonstructural-protein 5a (NS5a). SOF/DCV and SOF/LDP are available in the market as combination tablets. The virtual screening of anti-viral drugs has demonstrated that SOF, DCV, and LDP have a strong binding score with RdRp of SARS-CoV-2 [11,12]. As of now, multiple clinical studies of SOF/DCV with a limited sample size have been conducted on COVID-19 patients in early 2020, and they have shown encouraging results. The group of COVID-19 patients treated with the SOF/DCV combination has demonstrated decreased mortality and rapid clinical recovery compared to the standard of care (SOC) [4,10,13,14]. SOF and LDP have also been evaluated in an open-label study where the combination was found to have a shorter time to clinical response when added to the SOC [9,15]. These clinical trials have mainly examined the effect of these combinations on the management of fever, headache, duration of hospital stay, probability of ICU admission, mechanical ventilation, and mortality rate. However, these studies have not documented and discussed the anti-viral activity of combination agents. Though the inflammatory response induced by SARS-CoV-2 is a primary cause of disease severity as well as prolonged morbidity, anti-inflammatory agents have the primary role in its management. Therefore, it is essential to investigate the effect of drug molecules on reducing or modulating viral replication in COVID-19 patients.

This study reports the comprehensive evaluation of SOF, DCV, and LDP on RdRp of SARS-CoV-2 by computational approaches such as interaction analysis and molecular dynamic simulations. The activity of these molecules was assessed in therapeutic mode by in vitro anti-viral assay in the Vero E6 cells. Subsequently, we went ahead with a clinical trial to evaluate the safety and efficacy of SOF/DCV and SOF/LDP combinations on COVID-19 patients.

## 2. Materials and Methods

### 2.1. Molecular Dynamic Simulation of RdRp in Complex with Anti-HCV Drugs

The crystal structure of ligand-bound RdRp (PDB: 7BTF) was downloaded from the RCSB Protein DataBank [8]. The ligand, water molecules, and other heteroatoms were removed from these structures. Grid was set around the RNA binding pocket of RdRp (dimension: 34 × 34 × 36 Å) using the AutoGrid program of AutoDock Tools [16]. The structures of remdesivir (RDV), LDP, DCV and SOF were downloaded from PubChem [https://pubchem.ncbi.nlm.nih.gov/ (accessed on 12 November 2022)]. Three-dimensional conformations were checked for stereochemical properties and then converted to *.pdbqt format using Autodock Tools [16]. The docking of RdRp with four selected ligands was performed using AutoDock Vina-based Lamarckian Genetic Algorithm (LGA) parameter using ten runs criteria [17].

Molecular dynamic (MD) simulation analysis was performed to obtain further details about intermolecular interaction, ligand binding stability, and target conformational changes upon ligand binding to RdRp. The docked complexes of RdRp with selected ligands were subjected to MD simulation. These simulations were performed with GROMACS 5.1.4 package (GROMACS User Manual version 5.1.4) using the CGenFF and Charmm36-march 2019 force fields for ligands and proteins, respectively [18]. TIP3P water molecules surrounded all the protein atoms. The net charge on the system was made zero by adding sodium counterions. The system was energy minimized using the steepest descent algorithm for around 1000 steps and equilibrated NVT (constant number, volume and temperature) and NPT (constant number, pressure and temperature) conditions. The velocity-rescaling algorithm and Parrinello–Rahman pressure coupling maintained system temperature (300 K) and pressure at 1 bar, respectively. MD simulations for all the complexes were run for 50 ns each. Visual Molecular Dynamics (VMD) was used to analyze the trajectories. Root Mean Square Deviation (RMSD) calculations for protein backbone and ligands were performed using Standard GROMACS tools over the complete simulation range. Details of methods and tools is as described earlier [19].

### 2.2. Cytotoxicity Assay for Selected Anti-HCV Molecules

African green monkey kidney cells (Vero cells, subtype E6) were used to evaluate the cytotoxicity of the drug molecules. Vero E6 cells were seeded in the 96 well plates at 80% confluency and treated with various concentrations of SOF, DCV, LDP and RDV as a positive control. The SOF/DCV and SOF/LDP combinations were added in the *w*/*w* ratio 1:0.15 and 1:0.23, respectively. MTT (3-[4,5-dimethylthiazol-2-yl]-2,5 diphenyl tetrazolium bromide) cell assay kit (Himedia, Mumbai, India) was used to perform cytotoxicity checked after 48 h of treatment to assess the viability of cells in the presence of drug molecules. Details about the material and procedure were described earlier [20]. Each concentration was assayed in triplicates, and the percentage of cell viability was calculated with respect to vehicle control.

### 2.3. Anti-SARS-CoV-2 Activity Assay

Vero E6 cells were seeded in 96 well plates at 80% confluency and were infected with SARS-CoV-2 (IND-ILS01/2020) isolate at a multiplicity of infection (MOI) of 0.1 for two hours. Later, the inoculum was aspirated, and fresh media containing different concentrations of the SOF, DCV, LDP, RDV, SOF/DCV, and SOF/LDP were added to the cells. The anti-viral activity of each concentration was assayed in triplicates. The viral RNA was isolated from the supernatant after 24 h of infection, and the SARS-CoV-2 viral load was determined by qRT-PCR as described previously [21]. RNA isolation was done for cell samples TANBead Maelstrom 4800 as per standard procedure. Further, this RNA was used for cDNA using random hexamers (TaKaRa PrimeScript 1st strand cDNA synthesis kit, Kusatsu) as mentioned earlier [22]. qRT-PCR was performed on synthesized cDNA using SYBR Green (Eurogentec) with primer specific to nucleocapsid gene (FP: GTAACACAAGCTTTCGGCAG and RP: GTGTGACTTCCATGCCAATG). SARS-CoV-2 N gene was used for generating the standard curve and using samples corresponding Ct values percentage of copy number/mL. The percentage reduction in viral loads was plotted compared to vehicle-treated controls.

### 2.4. Clinical Trial

#### 2.4.1. Study Design and Participants

A triple-arm, parallel-group, hybrid, individually randomized, controlled clinical study was conducted at a public tertiary care teaching university hospital in northern India. Ethical approval was obtained from the institutional ethics committee of the hospital vide letter number IEC/20/213 dated 14 August 2020 to conduct the clinical trial. Patients were recruited from April 2021 to September 2021 for the study. The clinical trial was prospectively registered on CTRI on 24 August 2020. All principles of good clinical practice and the WMS declaration of Helsinki were followed [23].

#### 2.4.2. Study Population

All adult patients over the age of 18 years of either sex with confirmed qRT-PCR positive reports for SARS-CoV-2 and symptom duration of 7 days or less were eligible for participation in the study. At the time of the clinical study, guidelines mandated the use of corticosteroids (dexamethasone) in subjects with apparent pulmonary involvement, as suggested by moderate to severe COVID-19 disease categorization. The use of anti-inflammatory drugs is known to hamper viral clearance, as reflected by the persistence of positive nasopharyngeal swabs; hence, we planned to enroll only mild cases who were not on any immunosuppressive therapy [24].

The patients with moderate to severe COVID-19 disease, pregnant or breast-feeding women, known severe renal impairment, and subjects not consenting to the study were excluded. Additionally, patients with any known hypersensitivity to the study drugs, elevated levels of alanine aminotransferase, or aspartate aminotransferase more than five times the upper limit of normal were not included in the current study. Additionally, patients on immunosuppressive therapy were excluded. The majority of the Indian population was unvaccinated for COVID-19 and uninfected by SARS-CoV-2; hence, these variables were not included in the list of inclusion or exclusion criteria.

#### 2.4.3. Study Procedure

All patients enrolled in the study were subjected to relevant baseline clinical and laboratory examination, and their demographic details and baseline parameters were recorded. All eligible patients were divided into three groups. Group 1 received standard of care (SOC) treatment as per hospital protocol, which was at the time paracetamol 500 mg on a needed basis. Groups 2 and 3 were administered a fixed-dose combination of SOF/LDP (400 mg:LDP 90 mg) and SOF/DCV (400 mg:60 mg) once daily for 14 days in addition to standard treatment, respectively. Patients were randomly allocated 1:2:2 (groups 1:2:3) to either of the three groups. The serial number was assigned to the patients by the study coordinator and was linked to a computer-generated randomization list assigning the treatment regimens. After providing detailed information about the study, a written consent form was obtained from all subjects. As far as blinding was concerned, the study coordinator, drug dispenser and the participant were aware of the randomization between group 1 versus group 2 or 3, but the treatment allocation between groups 2 and 3 was not disclosed, hence making it a study with a hybrid design.

#### 2.4.4. Study Endpoints

The primary endpoint of the study was the proportion of patients achieving qRT-PCR negativity by day 7 in each arm. Secondary endpoints included hospitalization rates, percentage of patients experiencing a worsening of disease severity, mortality and treatment-induced adverse effects. For disease severity, ICMR guidelines and WHO recommendations for the minimum common outcome measure set for COVID-19 clinical research were followed [25,26]. Participants were evaluated for cough, sore throat, breathlessness, diarrhea, chest pain, body ache, nasal discharge, fever, headache, fatigue, ECG, and SPO2 at the time of enrollment to the completion of the trial. The cycle threshold for SARS-CoV-2 qRT-PCR was measured using nasal/oropharyngeal swabs at days 0, 3, 7, and 14 as per the standard recommended procedure [27]. The inflammatory markers such as CRP and ferritin (at days 0, 3, 7 and 14 for CRP and at days 0 and 14 for ferritin) were also measured, whereas D-dimer levels were measured at days 0 and 7. Additionally, the pulse rate was monitored on days 0 and 14. Study participants were telephonically monitored for disease progression based on WHO ordinal scale for COVID-19 disease severity. As an exploratory endpoint, we also conducted next-generation sequencing on all participant samples to evaluate the differences in the efficacy of the drugs based on the SARS-CoV-2 variants.

#### 2.4.5. Statistical Analysis

For sample size calculation, we assumed 50% negativity of qRT-PCR in the intervention arm would be clinically meaningful for repurposing, compared to the 20% (known spontaneous) negativity in the SOC arm, we calculated that a total of 145 subjects would provide 80% power to detect a hazard ratio of 2.5 with a two-sided significance level of 0.05 anticipating an attrition rate of 20%. We planned to assume higher attrition and enroll 175 subjects at the suggestion of the ethics committee. Anti-viral activity assays were performed in triplicates, and data were expressed as mean ± SD. Categorical variables were compared using the chi-square test, whereas continuous variables were compared among the three groups using the ANOVA test. The longitudinal change in the levels of CRP, D-dimer, ferritin, and pulse rate in SOC, SOF/DCV and SOF/LDP independently over time was analyzed by a two-tailed, paired t-test using GraphPad Prism v5.0 (GraphPad Software, San Diego, CA, USA). The rest of the evaluation was performed on SPSS^®^ (Statistical package for social sciences) Version 26 by IBM. A *p*-value < 0.05 was considered statistically significant.

## 3. Results

### 3.1. Drugs under Investigation Showed Stable and Competitive Binding with RdRp

Docking of RDV, DCV, LDP, and SOF with RdRP showed a strong binding score and affinity (Appendix A). The MD simulation and interaction analysis were extended to the RdRp complex with selected ligands. All ligands showed stable binding inside the RNA binding pocket of RdRp post 50 ns simulation. The RMSD graphs for RdRp protein exhibited a similar pattern for all complexes. The RMSD for protein backbone, DCV, and LDP ranges from 0.15 to 0.4 nm, indicating stable complex formation (Figure 1A,B). The potential energies of the complexes were stable after simulating for 50 ns. Molecular interaction analysis exhibited that DCV made polar contact with Lys593, and it was supplemented with Pi-Alkyl interaction between ligand and RdRp residues (Phe594, Cys813, Trp598 and Lys593) (Figure 1C).

In the case of LDP, Arg858, Ala685, and Ala688 were involved in establishing either Pi-cation or Pi-Alkyl interaction with the ligand groups (Figure 1D). RDV and SOF have known RdRp inhibitors; hence, they displayed higher binding scores and more density of intermolecular interactions in binding pocket residues and functional groups of the ligands [8]. Throughout the simulation of RdRp in complex with SOF and RDV, the RdRp protein backbone and ligands showed a minimal deviation in RMSD of 0.1 to 0.4 nm, and all the complexes depicted minimum potential energy at the end of 50 ns (Appendix A). The validation of the binding of these selected ligands and RdRp suggested that they can be explored for repositioning against SARS-CoV-2.

### 3.2. The Combination of SOF/DCV and SOF/LDP Demonstrated Synergistic Anti-SARS-CoV-2 Activity

The cytotoxicity assay was performed for single and a combination of drugs for 48 h incubation. The combination ratios of SOF/DCV (1:0.15) and SOF/LDP (1:0.23) were considered based on their fixed-dose concentration in the FDA-approved tablets [9,28]. The maximum non-toxic dose of individual and combinatory drugs was obtained at 200 µM with more than 90% cell viability (Figure 2). The drugs and their combinations were further evaluated for SARS-CoV-2 anti-viral activity after 24 h of virus infection at an MOI of 0.1 to the Vero E6 cells. The SOF and LDP individually showed the IC_50_ of SARS-CoV-2 around 42 µM and 96 µM, respectively. In contrast, DCV demonstrated IC_50_ around 7 µM, close to the IC_50_ value of positive control, RDV, around 4 µM (Figure 2). Interestingly, the combination SOF/DCV and SOF/LDP decreased the viral load more than the individual drug treatment with IC_50,_ around 1.8 µM and 2.0 µM, respectively (Figure 2).

### 3.3. Safety and Efficacy of SOF/DCV and SOF/LDP in The Management of COVID-19 Patients

Over a period of 6 months, 198 subjects were screened, and 183 patients were enrolled in the clinical trial. This period coincides with the second wave of COVID-19 in India. The most common reason for the failure of screening was the presence of uncontrolled co-morbidities or baseline hypoxia (Figure 3). Out of these, 43 patients were randomly assigned to the SOC group, and 76 and 64 patients were allocated to SOF/DCV and SOF/LDP groups, respectively (Figure 3). The mean age of the recruited patients was 34.68 ± 8.8 years, of which 114 (62%) were male versus 69 (38%) were female subjects. The foremost common symptoms in the enrolled patients were sore throat (62%), cough (63%), fever (62%) and fatigue (48%). Details of the baseline characteristics of participants are mentioned in Table 1.

There was no statistically significant difference between SOC and SOF/DCV, as well as SOC and SOF/LDP groups across a range of clinical and laboratory features at the baseline except CRP, which was higher in both SOF/DCV (*p* = 0.02) and SOF/LDP (*p* = 0.03) groups (Table 2). As a primary outcome of the study, qRT-PCR negativity for SARS-CoV-2 was observed in 91% of patients in SOC, whereas 87 and 88% in SOF/DCV and SOF/LDP, respectively, on the seventh day. The percentage of patients with undetectable SARS-CoV-2 RNA was comparable in all three groups (SOC vs. SOF/DCV, *p* = 0.53; SOC vs. SOF/LDP, *p* = 0.6). All samples were observed to be negative for the qRT-PCR test on the 14th day. The treatment points mentioned for this study in the international clinical trials registry platform were for 28 days. However, all the cases in the trial were COVID-19-negative by the 14th day and did not display any adverse effects clinically and in hematology. Hence, repeat tests for the primary outcome (nasopharyngeal qRT-PCR) were not repeated after the 14th day.

The secondary outcomes of the study indicated that none of the patients had any worsening in severity scale as assessed by mild/moderate/severe classification or as per WHO ordinal scale. None of the patients were hospitalized, and no mortality was observed in this clinical study. The post hoc exploratory analysis demonstrated that the levels of inflammatory markers, CRP and ferritin, were similar in all three groups on the 14th day. The D-dimer levels on the seventh day were significantly reduced in SOF/DCV group, as compared to SOC (*p* = 0.02); on the other hand, the levels were similar in SOC and SOF/LDP groups (*p* = 0.87). There was also a significant reduction in the pulse rate in SOF/DCV and SOF/LDP groups, as compared to SOC on the 14th day (Table 3). The change in CRP levels on the 14th day was significantly reduced to around 63 and 72% in SOF/DCV (*p* < 0.0001) and SOF/LDP (*p* < 0.0001) groups, respectively, as compared to an 18% non-significant reduction in SOC (*p* = 0.55) from the baseline. There was no significant difference in the ferritin level in all three groups between the baseline and the 14th-day endpoint. The D-dimer levels were significantly decreased to around 30% in SOF/DCV (*p* = 0.016) on the seventh day, whereas SOC (*p* = 0.31) and SOF/LDV (*p* = 0.31) demonstrated around 11% non-significant reduction. The pulse rate was significantly lowered in both SOF/DCV (*p* < 0.0001) and SOF/LDP (*p* < 0.0001) groups, but it was not reduced significantly in SOC (*p* = 0.9) (Figure 4). Time to symptomatic resolution was similar in the three arms. For safety endpoints assessment, biochemical parameters were similar in the three groups. All Samples underwent NGS and were found to have delta variant of SARS-CoV-2 (data not shown or included in this report).

## 4. Discussion

The index study was focused on the comprehensive understanding of anti-HCV drugs, mainly SOF, DCV and LDP, as candidate drugs for COVID-19 treatment by using in silico, in vitro, and clinical studies. Despite the demonstration of efficacy and anti-SARS-CoV-2 activity in pre-clinical benchside models (molecular dynamic simulation and anti-viral assay in the cells), both combinations failed to demonstrate any clinically significant viral clearance on day 7 when compared with the control arm.

SOF is an approved anti-HCV drug that acts as a substrate for HCV-RdRp [29]. It inhibits viral replication by incorporating into the newly synthesized viral RNA and thereby terminates its synthesis [15]. DCV and LDP are inhibitors of NS5a protein which have a critical role in viral RNA replication, protein phosphorylation and cell signaling [30,31]. The RdRp and NS5a proteins of both HCV and SARS-CoV-2 share similarities; hence, all these three drugs could be suitable candidates for COVID-19 treatment [32]. The SOF/DCV and SOF/LDP are approved combinations for HCV treatment. In the current study, systematic work was performed by in silico computational analysis for the binding affinity of drugs to the virus target protein, in vitro anti-viral activity, clinical trials and data analysis for the detailed investigation of SOF/DCV and SOF/LDP anti-viral combinations for COVID-19 treatment. The in silico MD simulation analysis demonstrated that these molecules have stable and competitive binding affinity to RdRp of SARS-CoV-2. Our results of MD simulation of SOF with SARS-CoV-2 RdRp corroborated previous reports suggesting that it has stable binding within the target protein’s active site and its binding energies are comparable to that of RDV [33,34]. Similarly, LDP has also been reported to have maintained a stable conformation with RdRp [35,36]. Although DCV has been shown to have a strong binding affinity molecular docking analysis, our study has validated its stable binding by MD simulation [37]. Furthermore, in vitro anti-viral assay in Vero E6 cells substantiated the inhibition of SARS-CoV-2 by the SOF/DCV and SOF/LDP combination. The presence of DCV and LDP increased the inhibitory potential of SOF by almost 20-fold. Thus, the combinations displayed better in vitro activity and IC_50_ than the individual drug molecules and the standard RDV. It has been previously shown by an enzymatic assay that SOF is a competitive inhibitor and chain terminator of SARS-CoV-2 RdRp [29]. The in vitro studies have also confirmed our findings that the sub-optimal concentration of DCV increases the potency of SOF, suggesting the synergistic effect of these molecules [38]. The anti-viral activity of the individual LDP has been reported in Vero E6 cells, but our study proved that LDP, similar to DCV, also exhibits a synergistic effect with SOF and increases its SARS-CoV-2 inhibition potential [35].

As these molecules indicated promising in silico and in vitro results, but the clinical study, which was extended to analyze the safety and efficacy of these drugs on the COVID-19 patients, failed to demonstrate any significant difference in viral load negativity at day 7 when compared to the standard of care. Nevertheless, the CRP level and heart rates were significantly reduced in both groups compared to SOC. The level of D-dimer decreased in SOF/DCV but not significantly in SOF/LDP and SOC. In contrast, the ferritin levels were comparable in all three groups. Many anti-viral molecules have been investigated for COVID-19 treatment. RDV has shown promising results in some clinical trials, but the results are not consistent [2,39]. Additionally, RDV needs to be administered intravenously; hence, it cannot be given to mild patients in an outpatient department (OPD) setting. Therefore, if oral treatment is initiated early in the onset of the disease, it may minimize the chances of hospitalization of the patients. Favipiravir, which has been demonstrated to have antiviral activity against hepatitis A virus, has also demonstrated doubtful clinical recovery in pilot studies, requiring more extensive trials to confirm these results [5,40]. SOF/DCV has been found to increase the rate of clinical response in OPD as well as hospitalized COVID-19 patients in mild to moderate cases [4,10,13,14]. Contrary to these reports, a recent clinical trial did not show any significant effect of SOF/DCV on the rate of hospital discharge or mortality in hospitalized COVID-19 patients, the majority of whom were from the severe disease category [28]. SOF/LDP has also shown improved clinical response in COVID-19 patients compared to SOC in previous studies [15]. A major limitation in these studies has been patient selection. All of these studies have included subjects who required oxygen as well as receiving concomitant corticosteroids for the inflammatory phase of COVID-19 disease. Given that viral persistence has little to do with the inflammatory ARDS phase, testing anti_HCV drugs for their anti-SARS-CoV-2 clinical activity will have significant confounders. The same has been demonstrated in a case report in recurrent COVID-19 infections despite being treated with direct acting antiviral [41]. To overcome this, we included only mild cases who were not on any immunomodulatory therapy. Our study did not demonstrate any clinical activity despite the results of pre-clinical experiments, which reflect upon the significant discordance between benchside and bedside medicine. Several reasons contribute towards this outcome like- the difference in the plasma concentrations and doses used in cytotoxic models, the level of penetration to the mucosal surface where the virus is mainly active, the role of other organs which influence the pharmacokinetics of the drugs and finally the genetic factors along with co-morbidities which impact the natural course of the disease.

In conclusion, this study has illustrated that the SOF, DCV, and LDP molecules stably bind to the RdRp target of SARS-CoV-2. Furthermore, the in vitro analysis showed that the SOF/DCV and SOF/LDP combinations also displayed better anti-viral activity than the individual molecules suggesting their synergistic effect. However, these combinations did not show a significant difference in qRT-PCR results as compared to SOC, although the longitudinal analysis revealed a reduction in the inflammatory markers.

## Figures and Tables

**Figure 1 vaccines-11-00515-f001:**
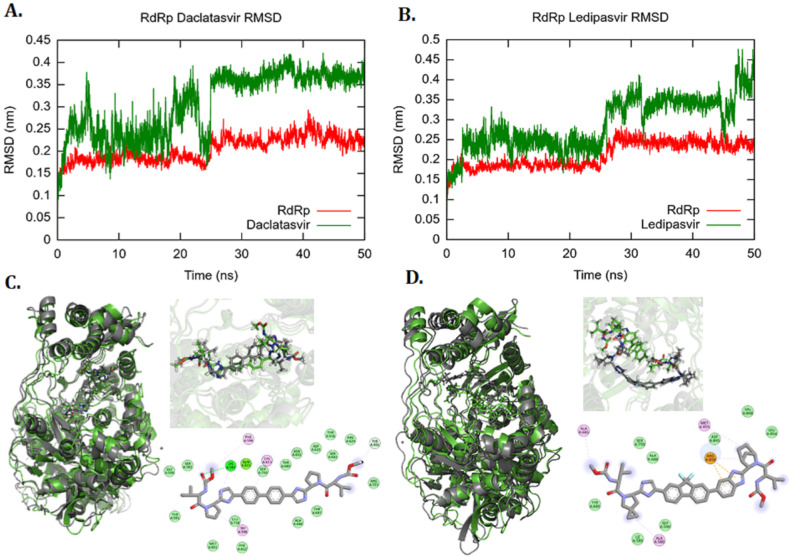
Molecular interaction analysis of simulated RdRp-ligand complexes RMSD plot of the (**A**) backbone of RdRP (Red) and DCV (Green) and (**B**) RdRp backbone (Red) and LDP (Green) for 50 ns MD simulations. Structural comparison and illustration of molecular interaction of (**C**) RdRp-DCV complex (**D**) RdRp-LDP complex, before and after 50 ns simulations. (RMSD: root mean square deviation; RdRp: RNA dependent RNA polymerase).

**Figure 2 vaccines-11-00515-f002:**
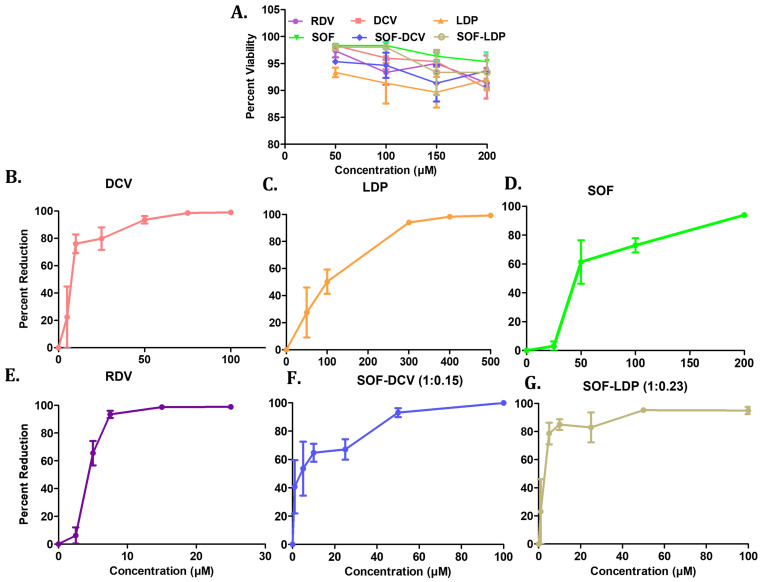
(**A**) Cytotoxicity of drug molecules was checked using MTT assay. Anti-viral activity of (**B**) DCV (**C**) LDP (**D**) SOF (**E**) RDV (**F**) SOF/DCV in 1:0.15 ratio and (**G**) SOF/LDP 1:0.23 ratio on Vero E6 cells in 96 well plates at 80% confluency. Graphs are represented as the percentage reduction of viral loads by qRT-PCR in infected cells with and without drug treatment. MTT: dimethylthiazolyl diphenyltetrazolium bromide; DCV: Daclatasvir; LDP: Ledipasvir; SOF: Sofosbuvir; RDV: Remdesivir.

**Figure 3 vaccines-11-00515-f003:**
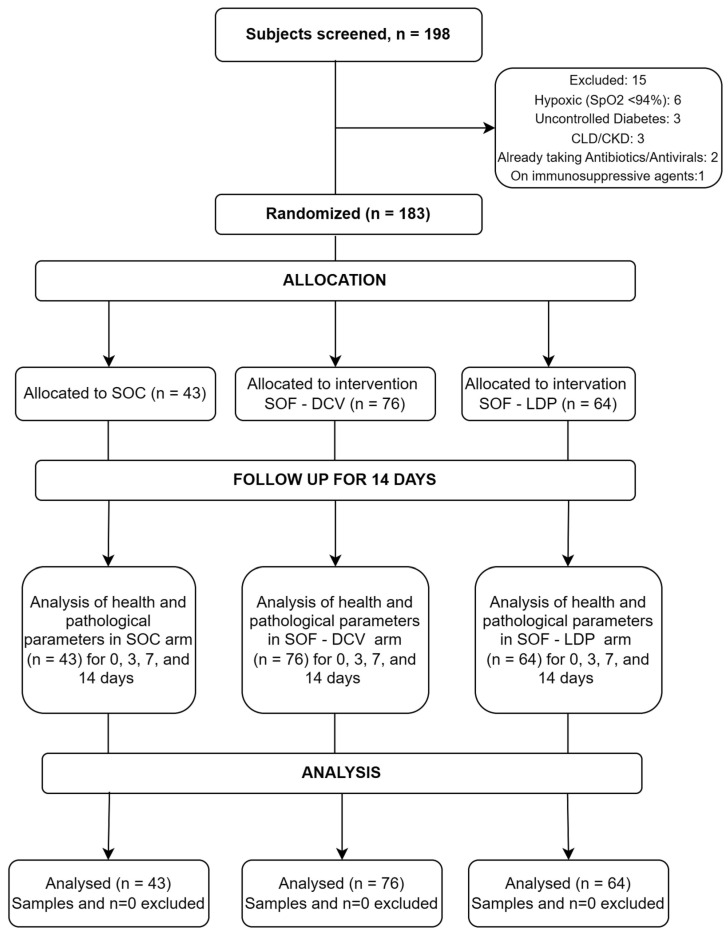
CONSORT Flowchart diagram for patient enrollment, randomization and follow-up. DCV: daclatasvir; LDP: Ledipasvir; SOC: standard of care; SOF, sofosbuvir: SpO_2_: Oxygen saturation as per pulse oximetry; CLD: Chronic liver disease; CKD: Chronic Kidney disease.

**Figure 4 vaccines-11-00515-f004:**
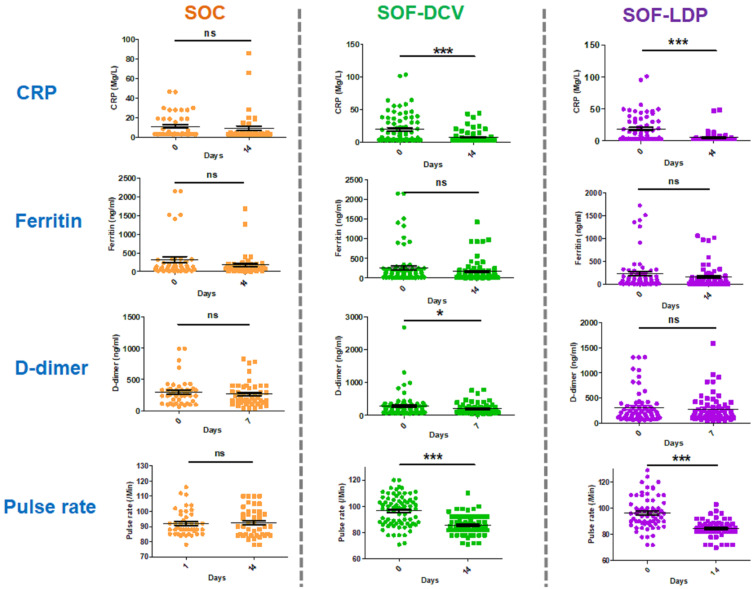
Longitudinal analysis of inflammatory parameters. CRP, Ferritin and cardiac health parameters, D-Dimer and pulse rate for SOF/DCV and SOF/LDP treatments compared to SOC. DCV: Daclatasvir; LDP: Ledipasvir; SOF: Sofosbuvir; SOC: Standard of care. *: *p* < 0.05; ***: *p* < 0.001.

**Table 1 vaccines-11-00515-t001:** Baseline characteristics of the study participants.

Character	SOC (n = 43)	SOF/DCV (n = 76)	SOF/LDP (n = 64)	Total (n = 183)
Age, mean (SD)	35 (9.1)	33.8 (8.2)	35.4 (9.2)	34.7 (8.8)
Male, n (%)	21 (49)	51 (67)	42 (66)	114 (62)
Female, n (%)	22 (51)	25 (33)	22 (34)	69 (38)
Sore throat, n (%)	26 (60)	48 (63)	39 (60)	113 (62)
Cough, n (%)	27 (63)	51 (67)	37 (57)	115 (63)
Breathlessness, n (%)	0 (0)	6 (8)	4 (6)	10 (5)
Diarrhea, n (%)	4 (9)	11 (14)	3 (5)	18 (10)
Chest Pain, n (%)	0 (0)	3 (4)	3 (5)	6 (3)
Body ache, n (%)	24 (56)	28 (37)	26 (41)	78 (43)
Nasal discharge, n (%)	13 (30)	17 (22)	8 (13)	38 (21)
Fever, n (%)	23 (53)	45 (59)	46 (72)	114 (62)
Headache, n (%)	14 (33)	22 (29)	16 (25)	52 (28)
Fatigue, n (%)	24 (56)	35 (46)	29 (45)	88 (48)

**Table 2 vaccines-11-00515-t002:** Baseline clinical and laboratory features of the study participants.

Clinical Characteristics	SOC, Mean (SD)	SOF/DCV, Mean (SD)	*p*-Value	SOC, Mean (SD)	SOF/LDP, Mean (SD)	*p*-Value
SPO_2_	98.05 (0.89)	98.21 (0.88)	0.33	98.05 (0.89)	97.95 (1.13)	0.65
Viral load by qRT-PCR	24.30 (3.80)	23.26 (3.91)	0.16	24.30 (3.80)	23.94 (3.87)	0.63
CRP (mg/L)	11.00 (12.29)	19.31 (22.53)	0.02	11.00 (12.29)	18.84 (21.93)	0.03
IL-6	16.88 (39.95)	8.37 (29.34)	0.18	16.88 (39.95)	9.2 (33.03)	0.28
Ferritin (ng/mL)	328.67 (555.05)	261.42 (448.82)	0.48	328.67 (555.05)	240.43 (390.81)	0.34
D-dimer (ng/mL)	303.74 (215.28)	277.61 (356.56)	0.66	303.74 (215.28)	310.32 (325.78)	0.9
Haemoglobin (g/dL)	12.53 (1.93)	13.21 (1.88)	0.08	12.53 (1.93)	13.32 (1.84)	0.06
TLC (cells/mm^3^)	5595.34 (1687.90)	5860.53 (1885.81)	0.38	5595.34 (1687.90)	6167.18 (1936.70)	0.1
Platelets (lakh/mm^3^)	1.73 (0.47)	1.79 (0.69)	0.5	1.73 (0.47)	1.8 (1.33)	0.6

**Table 3 vaccines-11-00515-t003:** Clinical outcomes of the patients after treatment.

Clinical Characteristics	SOC, Mean (SD)	SOF/DCV, Mean (SD)	*p*-Value	SOC, Mean (SD)	SOF/LDP, Mean (SD)	*p*-Value
CRP (mg/L), 14th day	8.97 (16.13)	7.06 (8.93)	0.4	8.97 (16.13)	5.29 (8.06)	0.12
Ferritin (ng/mL), 14th day	179.80 (305.32)	165.82 (260.63)	0.79	179.80 (305.32)	165.30 (247.45)	0.78
D-dimer (ng/mL), 7th day	268.72 (196.32)	195.94 (147.92)	0.02	268.72 (196.32)	276.49 (277.15)	0.87
Pulse rate, 14th day	92.37 (9.46)	85.76 (7.00)	0	92.37 (9.46)	84.64 (6.24)	0

## Data Availability

Individual anonymized patient data can be requested from the corresponding author.

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
