# Peer review of "A Comprehensive Molecular and Clinical Investigation of Approved Anti-HCV Drugs Repurposing against SARS-CoV-2 Infection: A Glaring Gap between Benchside and Bedside Medicine"

_vaccines, 2023, doi:10.3390/vaccines11030515_

Round 1
Reviewer 1 Report
Bansode et al. reported that a comprehensive molecular and clinical investigation of approved anti-HCV drugs repurposing against SARS-CoV-2 infection: A Glaring Gap between Benchside and Bedside Medicine. This looks important in this area.
1. In Introduction section, please delete the following sentence “Unfortunately, none of the anti-viral molecules has displayed conspicuous efficacy COVID-19 treatment to date.” Authors should mention about Remdesivir, Ensitrelvir and Molnuprevir, and their efficacy for COVID-19.
2. In discussion section, add the recent work and discuss more: Ikegami C, et al. COVID-19 After Treatment With Direct-acting Antivirals for HCV Infection and Decompensated Cirrhosis: A Case Report. In Vivo. 2022 Jul-Aug;36(4):1986-1993. doi: 10.21873/invivo.12923. PMID: 35738621
3. Authors should refer the similar studies in the polymerase inhibitors of other viruses. eg) Sasaki-Tanaka R, et al. Favipiravir Inhibits Hepatitis A Virus Infection in Human Hepatocytes. Int J Mol Sci. 2022 Feb 27;23(5):2631. doi: 10.3390/ijms23052631. PMID: 35269774
Author Response
To the Reviewers.
Dear Ma’am/Sir
We are extremely thankful for your efforts in reviewing the paper and providing us with your well-thought comments. We have discussed this among all authors and have modified our manuscript extensively. All suggested changes and points have been incorporated and are highlighted as per track changes.
For the English revision, we used the Grammarly Software to objectively current the language.
We are hopeful that you might find the revised version of the manuscript suitable for publication.
Sincerely
Dhruva Chaudhry.
Reviewer 1.
Bansode et al. reported that a comprehensive molecular and clinical investigation of approved anti-HCV drugs repurposing against SARS-CoV-2 infection: A Glaring Gap between Benchside and Bedside Medicine. This looks important in this area.
Response: We are delighted to know that editor and reviewers find our work important for the field
In Introduction section, please delete the following sentence “Unfortunately, none of the anti-viral molecules has displayed conspicuous efficacy COVID-19 treatment to date.” Authors should mention about Remdesivir, Ensitrelvir and Molnuprevir, and their efficacy for COVID-19.
Response: Suggested correction has been done in revised manuscript (line 52 to 54 and 57 to 60), also relevant citations has been added.
- In discussion section, add the recent work and discuss more: Ikegami C, et al. COVID-19 After Treatment With Direct-acting Antivirals for HCV Infection and Decompensated Cirrhosis: A Case Report. In Vivo. 2022 Jul-Aug;36(4):1986-1993. doi: 10.21873/invivo.12923. PMID: 35738621
Response: A sentence discussing this report has been added in revised manuscript as follows-
‘The same has been demonstrated in a case report in recurrent COVID19 infections despite being treated with direct acting antiviral’. The citation for the same has also been included.
- Authors should refer the similar studies in the polymerase inhibitors of other viruses. eg) Sasaki-Tanaka R, et al. Favipiravir Inhibits Hepatitis A Virus Infection in Human Hepatocytes. Int J Mol Sci. 2022 Feb 27;23(5):2631. doi: 10.3390/ijms23052631. PMID: 35269774
Response: Sentence mentioning this report has been added in revised manuscript as follows-
‘Favipiravir, which has been demonstrated to have antiviral activity against hepatitis A virus, has also demonstrated doubtful clinical recovery in pilot studies, requiring more extensive trials to confirm these results’. Also suggested citation has been added in the manuscript.

Reviewer 2 Report
This study investigated the anti-SARS-CoV-2 potential of anti-HCV drugs like daclatasvir (DCV) or ledipasvir (LDP) in combination with sofosbuvir (SOF). The binding mode and affinity of these molecules with RNA-dependent-RNA-polymerase of SARS-CoV-2 were analyzed by computational analysis. In- vitro anti-SARS-CoV-2 activity depicted that SOF/DCV and SOF/LDP combination has IC50 of 1.8 and 2.0 μM, respectively. Furthermore, the clinical trial was conducted in 183 mild COVID-19 patients for 14 days to check the efficacy and safety of SOF/DCV and SOF/LDP compared to the standard of care (SOC) in a parallel group. The primary outcomes of this study suggested no significant difference in negativity after 3, 7, and 14 days in both treatments.
Several suggestions:
1. Line 49, please add a reference after [over 6.2 million deaths]; also in line 54, after [for COVID-19 treatment to date]; line 104, after [downloaded from PubChem]; line 150, [declaration of Helsinki were followed]; line 157, [The use of anti-inflammatory drugs is known to hamper viral clearance]; line 254, after [in the FDA-approved tablets]; line 336, after [as a substrate for HCV-RdRp].
2. Line 75, [RdRp] is suggested to written as [ORF1ab (RdRp)] at the first appearance.
3. Line 88, [modulating viral particles] is changed to [modulating viral replication] or others.
4. Lines 95-97, please check [Although qualitative and quantitative health parameters were analyzed to examine the inflammatory response and cardiac health of patients treated with SOF/DCV and SOF/LDP.]. I don’t know what it means.
5. Line 117, full names for [NVT] and [NPT]?
6. Line 129, the full name for MTT and the name of the commercial-available kit.
7. Line 135, which variant of SARS-CoV-2 was used?
8. Line 139, please write down the condition for qRT-PCR or the commercial-available kit. The primer sequences for the detection of nucleocapsid gene?
9. [2.4.4. Study endpoints], [RT-PCR] could not compare the viral load, please change to [qRT-PCR] for the entire manuscript. How to perform qRT-PCR here? Same as in vitro study in 2.3?
10. Lines 218-220, [2.4.6. role of the funding source] should not be written here.
11. Line 222, [in-silico, in-vitro and in-vitro]. The second [in vitro] should be [in vivo] or [clinical study].
12. Figure 2A, why there is a slope between 0-50 uM? Drugs are more toxic to the cells in the lower concentration?
13. Line 321, please add [data not shown] after [have delta variant of SARS-CoV2].
14. Line 333, [molecular dynamic simulation and cytotoxicity assay]. It is better to use [anti-viral assay in the cells] or other phrases than [cytotoxicity assay].
15. Lines 339-340, [The RdRp and NS5a proteins of both HCV and SARS-CoV-2 share similarities]. Please explain more. Why these three proteins are similar? To me, they are very different.
16. If the references are available, please compare the binding mode and/or affinity of ORF1ab (RdRp) of SARS-CoV-2 with these drugs [DCV, LDP, SOF] to the binding of the NS5A and NS5B of HCV with these drugs.
Author Response
Dear Ma’am/Sir
We are extremely thankful for your efforts in reviewing the paper and providing us with your well-thought comments. We have discussed this among all authors and have modified our manuscript extensively. All suggested changes and points have been incorporated and are highlighted as per track changes.
For the English revision, we used the Grammarly Software to objectively current the language.
We are hopeful that you might find the revised version of the manuscript suitable for publication.
Sincerely
Dhruva Chaudhry.

Round 2
Reviewer 1 Report
All queries have been addressed.